Transfer of intestinal bacterial components to mammary secretions in the cow

Young Wayne 1
Hine Brad C. 2 *
Wallace Olivia A.M. 2
Callaghan Megan 2
Bibiloni Rodrigo 1 ** rodrigo.bibiloni@arlafoods.com
1 Food Nutrition & Health Team, Food & Bio-based Products Group, AgResearch Ltd, Grasslands , Palmerston North , New Zealand
2 Dairy Foods Team, Food & Bio-based Products Group, AgResearch Ltd, Ruakura , Hamilton , New Zealand
Anger Martin
* Current affiliation: CSIRO Animal, Food & Health Sciences, Armidale, NSW, Australia

** Current affiliation: Arla Strategic Innovation Centre (ASIC), Cultures & Fermentation Team, Denmark

Electronic publication date: 2015 Apr 23
Publication date: 2015
Volume: 3
Electronic Location ID: e888
Received 2014 Nov 21; Accepted 2015 Mar 20
Copyright: © 2015 Young et al.
Copyright year: 2015
Copyright holder: Young et al.
License: This is an open access article distributed under the terms of the Creative Commons Attribution License, which permits unrestricted use, distribution, reproduction and adaptation in any medium and for any purpose provided that it is properly attributed. For attribution, the original author(s), title, publication source (PeerJ) and either DOI or URL of the article must be cited.
License URL: https://creativecommons.org/licenses/by/4.0/

Keywords: Bifidobacterium, White blood cells, Feces, Cow milk, High-throughput sequencing, Mammary gland

Funding: Waikato Medical Research Foundation #185 This work was funded in part by the Waikato Medical Research Foundation (Grant #185). The funders had no role in study design, data collection and analysis, decision to publish, or preparation of the manuscript.

==============================
Results from large multicentre epidemiological studies suggest an association between the consumption of raw milk and a reduced incidence of allergy and asthma in children. Although the underlying mechanisms for this association are yet to be confirmed, researchers have investigated whether bacteria or bacterial components that naturally occur in cow’s milk are responsible for modulating the immune system to reduce the risk of allergic diseases. Previous research in human and mice suggests that bacterial components derived from the maternal intestine are transported to breast milk through the bloodstream. The aim of our study was to assess whether a similar mechanism of bacterial trafficking could occur in the cow. Through the application of culture-independent methodology, we investigated the microbial composition and diversity of milk, blood and feces of healthy lactating cows. We found that a small number of bacterial OTUs belonging to the genera Ruminococcus and Bifidobacterium, and the Peptostreptococcaceae family were present in all three samples from the same individual animals. Although these results do not confirm the hypothesis that trafficking of intestinal bacteria into mammary secretions does occur in the cow, they support the existence of an endogenous entero-mammary pathway for some bacterial components during lactation in the cow. Further research is required to define the specific mechanisms by which gut bacteria are transported into the mammary gland of the cow, and the health implications of such bacteria being present in milk.

Introduction

Epidemiological studies have shown that growing up on a farm is associated with a lower risk of developing allergy (Braun-Fahrländer et al., 1999; Ehrenstein et al., 2000; Waser et al., 2004; Alfvén et al., 2006; Mutius & Vercelli, 2010) and that the consumption of raw milk is strongly correlated with this effect (Perkin & Strachan, 2006; Waser et al., 2007). The commercialisation of fresh raw (unpasteurised) milk remains a controversial issue (Massie, 2014; Astley, 2014), and although its consumption has been discouraged due to potential health risks associated with pathogens (Allerberger et al., 2003), there is a growing body of evidence suggesting that consumption of unprocessed milk decreases the risk of asthma in children (Riedler et al., 2001; Loss et al., 2011). The mechanism for this effect is not yet fully understood, but it may be related to bacteria or bacterial components in the milk that modulate the immune system and which are modified during milk processing (Gehring et al., 2008; Hodgkinson, McDonald & Hine, 2014).

Although rural families generally skim or heat raw milk before consumption, this milk differs in many aspects from commercially available milk. Whereas commercial milk is usually homogenised and pasteurised, milk obtained from the farm for local consumption is not processed in these ways, potentially resulting in a higher microbial load. Results from PASTURE, a large global study examining the effects of farm and urban living on asthma and allergy, showed no differences in endotoxin levels of raw farm and commercially processed milk (Lluis et al., 2014); however, samples were not analysed for other bacterial components or presence of viable organisms.

It has been reported that human breast milk is not completely sterile (Heikkilä & Saris, 2003; Martín et al., 2003; Beasley & Saris, 2004; Jiménez et al., 2008), and that bacterial components derived from the maternal intestine are transported to the lactating breast by phagocytic cells in the blood (Pérez et al., 2007). This fascinating observation suggests that milk acts as a source of natural inoculum provided by the mother for the breast-fed newborn that programs the neonatal immune system to respond appropriately when challenged with specific environmental and dietary bacterial antigens.

We hypothesise that a similar mechanism of bacterial trafficking from the intestinal tract to the mammary gland, previously reported in lactating mothers, also occurs in the cow, contributing to the bacterial components found in raw cow’s milk and facilitating bacterial imprinting of the neonatal immune system. The identification of bacteria or bacterial components in unprocessed cow’s milk thought to originate from the intestinal tract would support this hypothesis. Therefore we investigated the microbial composition of feces, milk leukocytes and blood leukocytes in lactating cows by pyrosequencing barcode-tagged 16S rRNA amplicons to identify bacterial groups common to all three microbial pools. Bacterial imprinting of the neonate via trafficking of gut bacteria into mammary secretions in the cow may also contribute to the protective effects that consuming raw cow’s milk has been shown to have on development of allergy in children. An understanding of the mechanisms by which gut bacteria in milk imprints the neonatal immune system may provide the foundation for new strategies on how to shape the intestinal microbiota of the infant to aid in the prevention of immune disorders.

Materials and Methods

Animals

All experimental procedures were approved by the Ruakura Animal Ethics Committee, AgResearch, New Zealand (application #12345). A total of 35 lactating cows (Friesian, Jersey or Friesian × Jersey cross) located at the Tokanui Dairy Research farm (Waikato, NZ) were screened for the presence of subclinical/clinical mastitis infection. Only cows identified as having a low pooled quarter somatic cell count (SCC), defined as <100,000 cells/mL, at previous routine herd testing (conducted by Livestock Improvement Corporation, Hamilton, NZ) were screened. As part of the screening procedure, individual quarter milk samples were collected from all cows and subjected to a rapid mastitis test (Shoof International Ltd., Cambridge, NZ) and somatic cell count. Those animals showing any signs of subclinical or clinical mastitis (rapid mastitis test [RMT] score >1 (gel formation detected) or SCC >200,000 cells/mL) in any quarter were not eligible for the study. A subset of 12 cows, not showing signs of subclinical or clinical mastitis as defined above, were subsequently randomly selected to participate in the study. These lactating cows (age: 2–4 years old; parity: 1–3) were 180 days in milk on average at the time samples were collected. Cows were on twice-daily milking. Five (5) aged-matched non-lactating, non-pregnant cows of the same breed and showing no signs of disease were selected as control animals for blood analysis. All cows were grazing and supplementary fed on a feed pad.

Sample collection

Udders were thoroughly cleaned and disinfected with 70% ethanol and methylated spirit-impregnated swabs (Meths Clear; Vetpak, Te Awamutu, New Zealand), paying particular attention to the teat end by polishing the teat orifice with the swabs. Afterwards, a 14-gauge Teflon cannula (Terumo Surflo, Costa Mesa, California, USA) was inserted into the teat canal and connected to a sterile sample container by a drip extension similar to that described by Vangroenweghe and colleagues (Vangroenweghe et al., 2001). The cannula had a bevelled edge which curled inwards to minimise any potential damage to the teat canal. A total of 800 mL of milk was collected from each cow and placed in the cold (4 °C) until transported chilled to the analytical laboratory in Hamilton, NZ. Immediately following the removal of the cannula, each quarter was sprayed with chlorhexidine teat spray (Teat X, Deosan, Waharoa, New Zealand). Somatic cell counts were performed frequently after the study to check for intramammary infections caused by the sampling procedure. Blood samples (approximately 450 mL) were collected via jugular venepuncture under local anaesthetic (lignocaine hydrochloride monohydrate 2%; Phoenix Pharm Ltd., Vethparm, New Zealand) and sedation (detomidine; Zoetis NZ Ltd., Mount Eden, New Zealand) into sterile blood collection bags (Fenwal Inc., Lake Zurich, Illinois, USA) containing citrate phosphate dextrose as anticoagulant and Adsol as a red cell nutrient solution. The bags were placed in the cold (4 °C) and transported chilled to the analytical lab in Fielding, NZ. Skin was disinfected with 70% ethanol before venepuncture. Fecal samples were collected directly from the rectum of each animal with a gloved hand, placed into a sterile container, and stored at −80 °C until further analysis in Palmerston North, NZ.

White blood cell separation from blood and milk samples

Milk somatic cells (MSCs) were isolated as previously described (Daley et al., 1991). Briefly, milk was allowed to warm to room temperature, and centrifuged in 200 ml aliquots, at 250 × g for 30 min. Milk cells were then washed with 80 mL phosphate buffered saline [PBS] (Dulbecco A; Oxoid Ltd., Basingstroke, UK) and centrifuged again for 15 min. Cells were then resuspended in 80 mL of PBS containing 100 µg/mL gentamycin sulphate (Boehringer Ingelheim Bioproducts, Ingelheim am Rhein, Rhineland-Palatinate) for 10 min to kill extracellular bacteria and in suspension. Following incubation, cell suspensions were spun for 10 min, and washed with 40 mL PBS. After another 10 min spin, cells were resuspended in 1 mL PBS, and stored at −80 °C until processed for DNA extraction as described below. All steps were completed at room temperature in sterile conditions in a laminar flow cabinet appropriate for cell culture work.

White blood cells (WBCs) were isolated as follows. Blood bags were centrifuged at 15 °C at low speed (2,000 × g for 5 min with no brake applied). Plasma supernatant, buffy coat layer and the upper layer of red cells were transferred to a platelet bag using a Fenwal plasma extractor (Baxter Healthcare, Deerfield, Illinois, USA), and centrifuged again (4,500 × g for 7 min). Supernatant was again removed with the manual extractor and discarded leaving approximately 80 mL of pelleted cells in a minimal volume of plasma in the bag. The centrifuged cellular pellets containing WBCs were resuspended in the remaining plasma, treated with gentamycin as described above, washed and processed for bacterial DNA extraction as described below.

Skin swabs

After disinfection with ethanol but before milk collection, each teat and a defined area immediately around the teat were swabbed using Amies charcoal swabs (Raylab NZ Ltd., Kelston, New Zealand), and plated on Columbia sheep blood agar and McConkey agar plates (Fort Richard Laboratories Ltd., Otahuhu, New Zealand) to check for bacterial contamination. Plates were incubated at 37 °C for 24 h in aerobic conditions. One (1) Columbia sheep blood agar plate from each sample was also incubated at 37 °C for 48 h in anaerobic jars using an anaerobic GasPak generator (BBL Becton Dickinson, Franklin Lakes, New Jersey, USA) for facultative anaerobes.

Bacterial DNA extraction

Total DNA was extracted from 200 mg of fecal samples using NucleoSpin Soil kits (Macherey-Nagel GmbH, Düren, Germany) according to manufacturer’s instructions, but with the following modification. Fecal samples were diluted in 700 µL of NucleoSpin lysis buffer SL2 and 150 µL SX buffer, and homogenised using a FastPrep FP120 Cell Disrupter (Qbiogene Inc., Carlsbad, California, USA) set to speed 6.5 for 45 s prior to column purification of DNA. Milk and blood cells were pelleted by centrifugation and DNA extracted from the cell pellets using the same method described for fecal samples.

High-throughput sequencing

Isolated DNA was then used to amplify the V3–V5 regions of 16S ribosomal DNA, with universal bacterial primers (Claus et al., 2011) containing GS FLX adapter sequences, a unique 8 nucleotide ‘barcode’, and template specific sequences; forward primer 5′-CGTATCGCCTCCCTCGCGCCATCAGNNNNNNNNAGGCCAGCAGCCGCGGTAA-3′, and reverse primer 5′-CTATGCGCCTTGCCAGCCCGCTCAGGCCRRCACGAGCTGACGAC-3′, with ‘N’ indicating barcode nucleotides. Amplification reactions were completed on a MasterCycler ProS thermocycler (Eppendorf AG, Hamburg, Germany). Fecal DNA was amplified using the following conditions; 95 °C for 4 min, 25 cycles of (95 °C for 30 s; 49 °C for 30 s; 72 °C for 60 s) and 72 °C for 7 min. The PCR product size was 604 base pairs. Milk and blood cell DNA was amplified using the following PCR conditions; 95 °C for 4 min, 40 cycles of (95 °C for 30 s; 49 °C for 30 s; 72 °C for 60 s) and 72 °C for 7 min. Several dilutions of template DNA were made if the presence of PCR inhibitors was suspected. Samples were pooled and sent to the commercial sequencing facility (Macrogen Inc., Seoul, South Korea). To control for environmental contamination resulting from PCR with universal bacterial primers and high cycle numbers (40), negative controls without template DNA were also sequenced.

Sequence analysis

Sequences were processed using QIIME 1.7. Reads were quality filtered (quality score window >50) and assigned to corresponding samples according to barcode sequences using default values for minimum/maximum allowable length of read (200/1,000), allowed number of ambiguous reads (6), and allowable homopolymer length (6) (split_libaries.py -w 50 -b 8 -g -r -f). The resulting demultiplexed sequences were denoised and chimera checked using the Greengenes alignment as a database (release GG_13_5). Sequences identified as chimeric were removed from subsequent analyses. Sequences showing 97% or greater similarity were clustered into operational taxonomic units (OTUs) using the UCLUST method. Representative sequences were assigned taxonomies using the rdp method against the Greengenes GG_13_5 database (default 0.8 confidence threshold). Alpha diversity and OTU networks were generated using QIIME 1.7. Hierarchical clustering analysis of bacterial profiles was performed in R 3.0.2 (R Core Team, 2013) using Euclidean distances and complete linkage clustering.

Quantitative PCR

Bacterial DNA was amplified by quantitative PCR (qPCR) with the bacterial 16S rRNA gene primers F_Bact 1369 (5′-CGG TGA ATA CGT TCC CGG-3′) and R_Prok 1492 (5′-TAC GGC TAC CTT GTT ACG ACT T-3′) (Suzuki, Taylor & DeLong, 2000), using a Rotor-gene 6,000 thermocycler (Qiagen, Hilden, Germany). Samples were measured in duplicate using 10 µL reactions consisting of 1 µL DNA template, 0.25 µL forward primer (10 pmol/µL), 0.25 µL reverse primer (10 pmol/µL), 3.5 µL nuclease-free water, and 5 µL of KAPA SYBR® FAST Universal 2X qPCR Master Mix. Cycling conditions consisted of 95 °C for 3 min, followed by 40 cycles of (95 °C for 20 s, 60 °C for 30 s, and 72 °C for 30 s). Calculated concentrations (ng/µL) were normalized to extracted DNA concentrations. The amount of DNA detected was expressed as equivalent number of Escherichia coli genomes per ng of total DNA to provide an estimate of the numbers of bacteria present.

Statistical analyses

Ninety five percent confidence intervals for bacterial DNA quantities were obtained using R version 3.0.2 (R Core Team, 2013). Significance of differences between mean DNA concentrations was determined using the non-parametric Kruskal–Wallis analysis of variance in R, with P values <0.05 deemed to be significant. Power analysis indicated that results from 5 non-lactating, non-pregnant control cows, and at least 10 lactating cows would detect a difference of 0 vs. 65% at the 5% significance level with 80% power, with each cow categorised on whether it shows trafficking (whether at least one OTU is present in all three compartments (feces, blood and milk)) or not.

Results

Amplification and sequencing of bacterial DNA from the three biological pools

Bacterial DNA originating from MSCs and WBCs proved difficult to amplify using traditional PCR conditions involving 25–30 cycles, probably due to its low abundance; thus, we increased the number of PCR cycles to 40. This practice, however, might lead to false positive results. Therefore, we sequenced the negative controls (no DNA template), and filtered the sequenced samples to exclude the OTUs found in these control samples.

A total of 190,245 quality-checked bacterial 16S rRNA gene sequences were obtained by pyrosequencing from all tested animals (min = 1,276, max = 25,175; median = 4,369). The mean number (±SEM) of sequences obtained from feces (n = 11), MSCs (n = 12), and WBCs (n = 11) was 8,062 ± 2,097, 4,164 ± 510 and 3,765 ± 653, respectively. The mean read length was 281 bp (min = 200; max = 564). The number of operational taxonomic units (OTUs) at 97% similarity were 2,163 (min = 16; max = 983; median = 262), excluding all OTUs and sequences found in the negative control sample (15 OTUs).

Quantitative PCR analysis showed that mean amounts of bacterial DNA present per ng of DNA extracted from MSCs (6.51 × 10−3 pg (95% CI [2.87 × 10−3–1.02 × 10−2 pg])) were higher than from WBCs (1.79 × 10−3 pg (95% CI [9.07 × 10−4–2.66 × 10−3 pg])), which indicates higher numbers of bacteria present in MSCs compared to WBCs.

No bacterial DNA was recovered from blood originating from the aged-matched, non-lactating, non-pregnant control animals. Skin swabs collected from the teats after cleaning, but before milk collection did not show signs of viable bacteria in the culture conditions employed, ruling out any direct bacterial contamination from the skin during milk collection.

Microbial composition of the three biological pools

Analysis of DNA extracted from MSCs, WBCs and feces uncovered a small number of OTUs that were observed in all three biological samples from at least one cow (Table 1). Sequence assignment to the closest related taxa using the Greengenes GG_13_5 database indicated that sequences classified as Ruminococcus genus, Peptostreptococcaceae family, and Bifidobacterium genus were found concurrently in all three biological samples in a total of five, five and four cows, respectively. Members of 15 bacterial phyla were detected in the WBCs, whereas 22 bacterial phyla were represented in the MSCs. In comparison, fecal DNA contained representatives from only 13 phyla despite having the highest overall diversity at the 0.97 OTU level.

Table 1 Bacterial OTUs (bacterial signatures) found in the three biological samples from one or more sampled animal.

Profiling was performed by pyrosequencing of bacterial DNA.

OTU ID	Phylum	Classification	Animal ID	
1,759, 851, 1,942	Firmicutes	Ruminococcus	3, 5, 7, 9, 12	
251, 1,805	Firmicutes	Peptostreptococcaceae	1, 5, 7, 8, 12	
2,052	Actinobacteria	Bifidobacterium	3, 5, 7, 8	
1,954, 980, 1,511, 883	Firmicutes	Lachnospiraceae	3, 5, 7	
119, 792	Firmicutes	Ruminococcaceae	1, 2, 7	
187, 259	Bacteroidetes	Bacteroidales	3, 5	
813	Bacteroidetes	Paludibacter	3, 5	
476, 478	Firmicutes	Sarcina	5, 12	
1,681	Actinobacteria	Agrococcus jenensis	5	
2,183	Actinobacteria	Microbacterium	8	
1,151	Actinobacteria	Nakamurellaceae	8	
589	Bacteroidetes	Bacteroidaceae 5-7N15	5	
1,188	Bacteroidetes	Parabacteroides	1	
2,149	Bacteroidetes	Bacteroidales S24-7	3	
67	Cyanobacteria	Streptophyta	12	
939	Firmicutes	Blautia	7	
1,983	Firmicutes	Clostridia	7	
1,533	Firmicutes	Clostridiaceae	7	
1,219	Firmicutes	Coprococcus	12	
1,053	Firmicutes	Turicibacter	2	
806	Planctomycetes	Pirellulaceae	5	
1,369	Proteobacteria	Agrobacterium	3	
1,834	Proteobacteria	Escherichia	7	
238	Tenericutes	Mollicutes RF39	5	

The microbial diversity of each environment, as shown by the mean Chao1 index ± SEM at a sampling depth of 1,095 sequences was 596 ± 17, 427 ± 59 and 107 ± 20 for feces, MSCs and WBCs respectively (Fig. 1). As expected, fecal samples had greater microbial diversity compared to blood and milk samples (P < 0.01). MSCs also had significantly greater microbial diversity than WBCs (P < 0.01).

Figure 1 Analysis of diversity within communities.

Curves indicate Chao1 index, a measure of community diversity, at each sampling depth, as shown on the X-axis. Error bars indicate SEM.

The profile of microbial sequences identified in the feces, MSCs and WBCs differed between the sampling sites as shown by the hierarchical cluster analysis of bacterial profiles (Fig. 2). The most prevalent bacterial groups detected in the feces included, as expected, members of the Firmicutes (F) and Bacteroidetes (B). In contrast, bacterial sequences in WBCs were predominantly from Mycoplasma (33.9%) and Streptophyta (24.1%). Bacterial profiles from milk more closely resembled that from feces (Figs. 2–4), with the most abundant groups including Staphylococcus (27.6%), Ruminococcus (7.2%), Peptostreptococcaceae (6.5%), Bifidobacterium (5.6%), Butyrivibrio (2.3%) (Table 2).

Figure 2 Cluster analysis of bacterial composition at genus level.

Heatmap showing hierarchical clustering analysis of bacterial composition profiles for the 50 most abundant genus level taxa as a proportion of total sequences for each sample. Coloured bar beneath upper dendrogram indicates sample environment; fecal (yellow), milk cell (blue), white blood cell (red). Taxa are indicated by row labels and individual animal and sample environment indicated by column labels; F (fecal), M (milk somatic cell), B (white blood cell), and numbers representing animal identification. Heatmap colour (blue to dark red) signify relative prevalence of each taxa across samples and green circles show absolute proportions for each taxa within a sample, with circle size proportional to taxa abundance.

Figure 3 Bar chart of bacterial composition at family level.

Stacked bar chart of the 100 most abundant family level taxa found in each of the faecal (n = 11), milk cell (n = 12), and white blood cell (n = 11) environments. Bars show mean bacterial proportions for each environment.

Figure 4 OTU network linking samples and OTUs found in each sample.

Sample types indicated by coloured shapes; fecal (yellow circle), milk cell (blue square), and white blood cell (red diamond). OTUs are shown by white dots and lines join OTU with sample which that OTU is found in. Lines are coloured according to sample type the OTUs are found in; fecal (yellow), milk cell (blue) and, white blood cell (red). Network arranged using a perfuse force directed layout where samples that share more OTUs are placed closer together.

Table 2 Abundance of bacterial taxa.

Profiling was performed by pyrosequencing of bacterial DNA. Taxonimic names listed correspond to the highest level identified (0.8 confidence) for each group of bacterial sequences detected.

Rank	Feces	White blood cells	Milk somatic cells	
1	(F) Ruminococcus (28.6%)	(T) Mycoplasma (33.9%)	(F) Staphylococcus (27.6%)	
2	(B) Bacteroidales (8.2%)	(C) Streptophyta (24.1%)	(F) Ruminococcus (7.2%)	
3	(F) Ruminococcaceae (6.5%)	Unclassified bacteria (13.4%)	(F) Peptostreptococcaceae (6.5%)	
4	(B) Bacteroidaceae 5-7N15 (6.0%)	(B) Prevotella (4.5%)	(A) Bifidobacterium (5.6%)	
5	(F) Unclassified Lachnospiraceae (4.0%)	(P) Stenotrophomonas (3.3%)	(F) Butyrivibrio (2.3%)	
6	(F) Lachnospiraceae (3.8%)	(P) Acinetobacter (1.1%)	(P) Stenotrophomonas (2.1%)	
7	(B) Paraprevotellaceae CF231 (3.5%)	(A) Micrococcus (0.8%)	(F) Ruminococcaceae (2.0%)	
8	(B) Rikenellaceae (3.4%)	(A) Kocuria (0.7%)	(A) Intrasporangiaceae (1.8%)	
Notes.

Phylum indicated by letter in parentheses.

F Firmicutes

B Bacteroidetes

T Tenericutes

C Cyanobacteria

P Proteobacteria

A Actinobacteria

Discussion and Conclusion

The general convention dictates that mammalian milk, including that of human and bovine origin, is at its origin free from microorganisms. According to the Food and Agriculture Organization of the United Nations, milk secreted into a cow’s udder is sterile (FAO, 1990). This theory, however, has been recently challenged. A few scientific studies using nucleic acid–based methodologies have started to reveal that colostrum and human breast milk contain microorganisms (Martín et al., 2003; Beasley & Saris, 2004; Pérez et al., 2007), becoming potential sources of bacterial exposure for the breast-fed newborn. Nevertheless, the origin of such microorganisms, as well as their health implications, are still a controversial issue. Although it is generally accepted that the presence of bacteria in milk can result from contamination with bacteria from the mother’s skin or the infant’s mouth, a newly proposed endogenous pathway to explain the origin of some milk bacteria is under debate. An entero-mammary pathway has been suggested by which selected bacteria from the maternal gastrointestinal microbiota reach the mammary secretions via the blood leukocytes (Martín et al., 2003; Pérez et al., 2007; Fernández et al., 2013; Rodríguez, 2014). We hypothesised that a similar mechanism of bacterial trafficking from the intestinal tract to the mammary gland, previously reported in lactating mothers and mice, also occurs in the cow, contributing to the bacterial components found in raw cow’s milk. Although our findings do not definitely prove our hypothesis, the presence of bacterial fragments in all three environments provides support for the occurrence of a trafficking mechanism of bacterial components from the intestinal tract to the mammary gland in the cow. Bearing in mind that the lactational physiology of humans and rodents is different to that of ruminants, the occurrence of this endogenous bacterial circulation would lead to new scientific insights into bovine physiology.

We employed an established culture-independent methodology to investigate the bacterial composition in the three biological compartments. Our results suggest that there are bacterial components belonging to the Ruminococcus genus, the Peptostreptococcaceae family, and the Bifidobacterium genus that can be found in common in feces, WBCs and MSCs from the same lactating cow. It is therefore reasonable to speculate that members of these bacterial groups may have been transferred from the gut to the mammary gland via circulating white blood cells. Pérez and colleagues observed DNA from Bifidobacterium longum in milk samples from lactating mothers which was also present in their blood and feces (Pérez et al., 2007). These researchers also found sequences from Bacteroides, Clostridium, and Eubacterium in human milk. Martín et al. (2003) have also isolated bifidobacterial species in human milk.

Two aspects in our approach require attention: namely, milk collection and PCR amplification. Common practise for milk sampling from cows is by hand stripping. When hand-stripping, it is very difficult to collect milk aseptically due to skin flakes, dust, and hair in the environment, which can all introduce bacterial contamination to the sample. Rather than hand-stripping, we used a catheter inserted into the teat canal which was connected to the sample container by a drip extension set that allowed milk collection by gravity. This procedure not only prevented the teat canal from being stretched or damaged (historical post-sampling SCC records suggest no intramammary infections were caused by the procedure, data not shown), but also avoided external microbial contamination of the milk. Although we cannot exclude that bacteria of skin origin or from the keratin lining could have colonised the milk duct of the cow’s udders and then transferred to the milk, we have designed our experiment to exclude any bacteria in suspension by collecting only white cells (the majority of the somatic cells in blood) by centrifugation. Additionally, these cells were treated with the bactericidal gentamycin, and consequently only internalised bacterial signatures or those coming from membrane bound bacteria were measured by high-throughput sequencing. Finally, the three laboratories located in different sites each processed only one type of the three biological samples (feces, blood and milk) which significantly reduces the possibility of introducing the same type of contamination to each of the samples.

Although environmental contaminants can become disproportionately represented when amplifying sequences from samples containing low copy numbers of bacterial DNA compared to non-target DNA, our results show that high-throughput sequencing is a useful method for assessing microbial composition in milk and blood. Despite the presence of contaminant sequences in the negative control samples subjected to 40 rounds of PCR amplification, the number of OTUs observed was substantially lower than that in blood and milk samples. Taxa found in common in all three environments were identified in a maximum of 5 cows out of the total tested animals. The variation in the number of taxa found in our study could be perhaps related to the stage of lactation, days in milk, or it could simply be from chance. Because we amplified very small quantities of bacterial DNA in the blood and milk, it may be also possible that bacterial species of interest present in some samples were not detected. Laser capture microdissection in combination with direct-captured cell PCR has recently been developed to identify microbial contaminants in milk, potentially representing a suitable tool to detect bacterial species present in low abundance (Bracke et al., 2004). Quantitative PCR analysis showed higher numbers of bacteria present in MSCs compared to WBCs, suggesting that cells with bacteria accumulated in the mammary gland as there were more bacteria per unit of mammalian DNA in the milk than there was in the blood.

Milk sample collection techniques and cell differentiation methodologies can influence milk cell differentiation results; however, it is generally accepted that macrophages are the predominant cell type in bovine milk from healthy glands (Miller, Paape & Fulton, 1991; Dosogne et al., 2003). Lymphocytes and neutrophils are also present along with a small percentage of detached epithelial cells which together make up the total somatic cell population. During inflammation, neutrophils are rapidly recruited to the mammary gland, becoming the predominant cell type and increasing somatic cell counts in inflamed quarters (Riollet, Rainard & Poutrel, 2000). It is noteworthy that although the methods used to screen cows for suitability to be used in the study suggested that selected animals were not harbouring an intramammary infection at the time of sampling, bacteriology analysis of quarter milk samples was not undertaken to confirm the absence of subclinical mastitis during the screening procedure. Nevertheless, the phagocytic macrophages were expected to be the predominant cell type present in milk from these animals. Macrophages in milk are thought to be derived from blood monocytes which exit the bloodstream, migrate across the epithelium and enter the mammary gland (Goldman & Goldblum, 1997). Macrophages play a key role in immune surveillance, acting as scavenger cells with the ability to recognise pathogens and initiate innate responses through the secretion of pro-inflammatory mediators. Following phagocytosis of antigens, some tissue macrophages differentiate into dendritic cells and migrate to draining lymph nodes where they interact with T-cells to induce antigen-specific acquired responses (Randolph et al., 1999). Macrophages can also function as antigen presenting cells, a subset of which are able to induce oral tolerance through interaction with CD103+ dendritic cells (Mazzini et al., 2014). Breast milk macrophages express certain dendritic cell surface markers, spontaneously produce granulocyte-macrophage colony-stimulating factor and have the unique ability to differentiate into dendritic cells when stimulated with interleukin-4 (Ichikawa et al., 2003). Combined, these findings suggest that breast milk macrophages exhibit characteristics consistent with that of partially differentiated dendritic cells, and that such cells may play a role in mediating T-cell dependant immune responses in the mammary gland. However, as the enhanced functionality displayed by milk macrophages is thought to be induced by phagocytosis of milk components following entry to the mammary gland, such findings do not provide an explanation as to the likely mechanisms by which gut bacteria and bacterial components are transported from the intestinal lumen to the mammary gland. The recirculation of lymphocytes between distant mucosal sites via the blood and lymphatic systems has been studied in several species. In humans and rodents, lymphoid cells in gut-associated lymphoid tissue (GALT) home to the mammary gland forming an entero-mammary link and contributing to what has been termed the ‘common mucosal system’ in which distant mucosal sites are linked via the migration of immune cells (Roux et al., 1977; Weisz-Carrington et al., 1979). In contrast to these findings, studies in cattle have demonstrated that migration of lymphoid cells between the gut and the mammary gland is limited, suggesting that the entero-mammary link is less functional in ruminants (Harp, Runnels & Pesch, 1988). More recently, it has been proposed that although the mononuclear phagocytes found in breast milk are largely derived from peripheral blood monocytes, a proportion of these mononuclear phagocytes are dendritic cell-like cells which arise in gut-associated lymphoid tissue, capture luminal microbiota and then transport these microbial components to the mammary gland. Such a mechanism is thought to exist to educate the neonatal immune system to recognise commensal-associated molecular patterns of bacteria and to respond to such bacteria appropriately (Donnet-Hughes et al., 2010). To the authors knowledge, the role of dendritic cells in trafficking bacterial components from the gut to the mammary gland in ruminants remains relatively unknown.

The possibility of bacterial trafficking from the gut to the mammary gland in the cow opens up interesting alternatives for probiotic use in the manipulation of the intestinal ecosystem for animal health. Ingested probiotics with the ability to get access to the mammary gland through the bloodstream could be employed to combat pathogenic microorganisms involved in the development of mastitis. Finally, further research is required to unequivocally link the biological activity of bacterial groups of interest in the development of allergy, in which case technological strategies in milk processing could be directed towards maintaining the integrity of such beneficial components.

The authors would like to thank Laura Mayall, Allison Cullum, and Taisekwa Chikazhe for their assistance with sampling at the farm, Harold Henderson for statistical advice, and Julie Cakebread, Henning Seedorf and Ian Colditz for reviewing the manuscript.

Additional Information and Declarations

Competing Interests

Author Contributions

Animal Ethics

DNA Sequence Deposition

Wayne Young, Olivia A.M. Wallace, and Megan M.R. Callaghan are employees of AgResearch Ltd., Brad C. Hine is an employee of Commonwealth Scientific and Industrial Research Organisation (CSIRO), and Rodrigo Bibiloni is an employee of Arla Foods Amba.

Wayne Young conceived and designed the experiments, performed the experiments, analyzed the data, contributed reagents/materials/analysis tools, wrote the paper, prepared figures and/or tables, reviewed drafts of the paper.

Brad C. Hine and Olivia A.M. Wallace conceived and designed the experiments, performed the experiments, analyzed the data, contributed reagents/materials/analysis tools, wrote the paper, reviewed drafts of the paper.

Megan Callaghan performed the experiments, analyzed the data, contributed reagents/materials/analysis tools, wrote the paper, reviewed drafts of the paper.

Rodrigo Bibiloni conceived and designed the experiments, performed the experiments, analyzed the data, wrote the paper, prepared figures and/or tables, reviewed drafts of the paper.

The following information was supplied relating to ethical approvals (i.e., approving body and any reference numbers):

Ruakura Animal Ethics Committee, AgResearch, New Zealand (application #12345).

The following information was supplied regarding the deposition of DNA sequences:

Raw sequence reads were deposited in the publicly accessible NCBI Sequence Read Archive database (accession number PRJNA280231; http://www.ncbi.nlm.nih.gov/bioproject/280231).

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
