# Peer review of "Transfer of intestinal bacterial components to mammary secretions in the cow"

_PeerJ, doi:10.7717/peerj.888_

## Round 0.1 · original submission · Major Revisions

The reviewers comments regarding better characterisation of the animals used in the study, details about sample collection and preparation, better description of sequencing techniques and analysis of the results are important. Please also consider rephrasing certain sections of the manuscript in order to harmonise the data and claims in the manuscript.

Reviewer 1 ·

Basic reporting

The manuscript is well-written and includes sufficient introduction and background. The consumption of raw cow’s milk contributes to the protective effect that has been shown to have on development of allergy in children. The mechanisms for this effect is not yet fully understood, but it may be related to bacteria or bacterial components in the milk that modulate the immune system. These bacterial components derived from the maternal intestine are transported to the lactating breast by phagocytic cells in the blood. The authors hypothesise that a similar mechanism also occurs in the cow, contributing to the bacterial components found in raw cow’s milk. The clarity of the research question is high.

No other specific comments

Experimental design

Milk leukocytes, blood leukocytes and fecal samples were collected from a subset of 12 cows that was randomly selected and five healthy aged-matched non-lactating, non-pregnant cows were selected as controls animals for blood analysis. The microbial composition of feces, milk leukocytes and blood leukocytes in lactating cows was determined by pyrosequencing barcode-tagged 16S rRNA amplicons to identify bacterial groups common to all three microbial pools.

No comments

Validity of the findings

Major comments:
- Line 203: The fact that bacterial taxa found in common in all three environments were identified in a maximum of 5 cows out of a total of 11 or 12 tested animals is intriguing. Authors should add comments about this, indicating hypotheses to justify why those bacteria (common to all environments) were not identified in the majority of lactating cows.

- Line 287-291: Authors proposed a possible mechanism to explain how gut bacteria could be transported to mammary glands, but this part should be completed by providing other explanations on the trafficking of gut bacteria to mammary secretions, as well as other methodologies than the isotopic tracing which could be used to confirm this hypothesis.

Additional comments

Minor comments:
- Line 140: Authors only mentioned quality score threshold but since pyrosequencing was used in this study, the following quality filters should be added: minimum/maximum allowable length of read, allowed number of ambiguous reads, and allowable homopolymer length.

- Line 145: Sequence identity threshold used to assign taxonomy (with Greengenes) to the OTU representative sequence should be provided.

- Line 170: Authors only indicated the mean number of sequences for each environment but we don’t know if this number varies widely between samples, and consequently, if authors had to equalize all samples to the smallest file size (= sample with the smallest number of sequences).

- Line 216: “We employed a widely accepted methodology published elsewhere to investigate…”. This sentence should be reformulated with more appropriate terms.

- Table 1: Legend should be completed indicating that listed phyla were identified in all three environments.

- Figure 2: Taxa labels should be larger to facilitate reading.

- Figure 4: Sample numbers are not really readable.

Reviewer 2 ·

Basic reporting

CENTRAL THEORIE OF LACTATION SCIENCES
Since decades, scientists accept that milk secretion in the bovine udder is occurring under sterile conditions and therefore, at its origin, milk does not contain bacteria. This has been well documented and confirmed by repeated studies. This is a reliable central theory in lactation sciences. According to the FAO, milk secreted into a cow's udder is sterile. Invariably it becomes contaminated during milking, cooling and storage. The nutrient composition of milk makes it a good medium for the growth of many different kinds of microorganisms. The cisterns of the gland can be infected with pathogens resulting in clinical or subclinical mastitis (http://www.fao.org/).

ROUTE OF MAMMARY INFECTIONS
A hematogenous infection of the mammary gland through the blood is extremely uncommon but might exceptionally occur (e.g. Brucella sp., Mycobacterium sp. Coxiella burnetii). It is suggested that, with the exception of some pathogens (e.g. Mycobacterium sp.), infections do not reach the mammary gland by a hematogenous route. Also Coxiella burnetii, the causative agent of Q-fever, may infect the udder through a hematogenous route (Richard K. Robinson [Ed.], 2002 Dairy Microbiology Handbook: The Microbiology of Milk and Milk Products. ISBN: 978-0-471-38596-7). Intramammary infection with Pastuerella sp. is rare and may be spread hematogenously in cows with respiratory infection (Mastitis Pathogen Factsheet #3, University of Minnesota College of Veterinary Medicine).

Over the past 20 years, phenotyping and genotyping methods have been used to study mastitis-causing bacteria at species and subspecies level. Genotyping methods used to characterize these pathogens range from PCR to whole genome sequencing. Typical bovine mastitis pathogens such as Escherichia coli and Klebsiella pneumoniae, Streptococcus agalactiae, Streptococcus uberis, and Staphylococcus aureus have been investigated with molecular methods (see review of Ruth Zadoks et al., 2011. J. Mammary Gland Biol. Neoplasia 16:357–372). All these pathogens breach the teat canal to infect teat and udder cisterns and induce an inflammatory reaction.

It is known that in the cow's udder tissue some non pathogenic bacteria may be present before the cow is suckled by its calf.

RATIONALE OF THE CURRENT STUDY
Sciences evolve continuously and demand for correction of old theories, and if necessary, replacing them with more accurate ones. Theories can be rejected on basis of new empiricism and (experimental) evidence, leading to new scientific insights. Observations and studies have to be testified permanently and rejected if proven to be false. Therefore studies as presented in the current manuscript should be encouraged.

In the introduction of the submitted manuscript the authors mention that research in human and mice suggests that "bacterial components" derived from the intestine are transported to the mammary gland through the bloodstream. The aim of the study was to assess whether a similar mechanism of "bacterial trafficking" occurs in the lactating cow. Therefore a "culture-independent methodology" was used to investigate the "microbial composition and diversity" of milk, blood and feces of healthy lactating cows.

RESULT AND CONCLUSION OF THE CURRENT STUDY
It was found that a small number of bacterial operational taxonomic units (OTUs) belonging to the genera Ruminococcus and Bifidobacterium, and the Peptostreptococcaceae family were present in all three samples from the same individual animals. The authors conclude that trafficking of gut bacteria into mammary secretions does occur in the cow.

GENERAL COMMENTS MADE BY THE REFEREE
The merit of the present study relies on the challenge of: 1) the central theory of lactation sciences (see FAO, milk secreted into a cow's udder is sterile). 2) The possibility of gut bacteria to breach the intestinal barrier with subsequent trafficking in the blood circulation, and eventually passage into the milk (by breaching the blood-milk barrier).

Conventional detection methods (the cornerstone of modern bacteriological methods) have been developed a century ago. Molecular detection methods are relatively new (approximately 20 years). Molecular detection can be used for (1) identification of an organism already isolated in pure culture, (2) rapid identification in a diagnostic setting from clinical sampels or (3) for the identification of an organism from unculturable specimens (e.g. culture negative samples, intracellular organisms; see Millar et al. 2007. Molecular Diagnostics of Medically Important Bacterial Infections. Curr. Issues Mol. Biol. 9: 21–40).

The main criticism and a major weakness of the study is that the results do "not prove" that bacteria as such breach the intestinal and blood milk barrier, and are transported in the systemic circulation. If this would have been the case one could conclude that bacterial sepsis is involved and hematogenous infection of the mammary gland. An alternative would be that one considers only intracellular surviving bacteria (e.g. staphylococci; see Proctor et al. 2006 Small colony variants: a pathogenic form of bacteria that facilitates persistent and recurrent infections. Nature Reviews Microbiology 4, 295-305).

This paper is not dealing with trafficking of gut bacteria over the circulation into mammary secretions. The paper is dealing with intracellular bacterial components (DNA, sequence analysis, qPCR with amplification of the bacterial 16S rRNA gene) in three distinct places in the cow: gut, blood en milk cells. From a scientific point of view this is in itself an interesting study that uses modern analytical techniques and there is no need to inflate the results with bombastic and speculative claims that have not been studied in the samples. There is no need to guess it in advance. Consider all consistent or inconsistent data and confront them to critical analysis.

The manuscript is suffering from speculation and overestimation of the results. It is missing focus and rigorous discussion (analytical and critical approach, missing pro's and contra's). There are flaws and wrong concepts. For example, the authors consider lactation physiology of human and rodents, identical to the physiology in bovines. Consider species specificity. The quality of the discussion is in sharp contrast with discussions of recent review papers of Rodrigo Bibiloni (e.g. 2010 & 2012). The scientific quality of the paper would enhance significantly if focused and if all speculative side information is omitted.

SPECIFIC COMMENTS MADE BY THE REFEREE
1) Title (change into more focused title?)
- very nice for a review paper but not covering the content of the current paper
2) Abstract has to be rephrased
- association between consumption of raw milk and reduced incidence of allergy in children: superfluous and misleading information because the paper did not study this.
- the results do not support transport of bacteria from gut to milk in the cow
- bifidobacteria claim is not a proof
3) Animals
- how were 35 animals screened for clinical/subclinical mastitis?
- cows 180 days in milk (6 months) -> what was the age of the cows (calving number)?
- what is a low bulk somatic cell count? SCC in tank? data needed.
- please use appropriate abbreviations for quater, individual and tank cells.
- what is a healthy cow; define.
- why were 12 cows selected ad random from 35 cows? Needs more explanation.
- details of non-lactating, non-pregnant animals needed (culled cows? age?)
- how were cows milked, milking frequency?, feeding and housing?
- how do you define samples negative for udder pathogens (on basis of how many samples?)?
4) Sample collection
- several devices have been described in literature to collect sterile milk samples (e.g. Vangroenweghe et al., 2001. Effect of milk sampling techniques on milk composition, bacterial contamination, viability and functions of resident cells in milk. Veterinary Research, BioMed Central, 2001, 32 (6), pp.565-579.
- cow's teat canal is 5 to 10 mm in length and a diameter of 0.4 to 1.6 mm. Some bacteria may survive in the keratin lining and secretions in the distal teat canal. During cannula insertion, these bacteria can be pushed into the teat cistern. How was this avoided?
5) WBC preparation
- milk: how can all steps be completed in absolute sterile conditions?
- for blood the term "blood leukocytes" is used under materials; under results the term "white blood cells" is used. This might be confusing. Use same terms.
6) Skin swabs teat end
- no comments
7) Bacterial DNA extraction/sequence analysis & qPCR
- what is the additive value of qPCR?
- comment, see also results
8) Kruskal-Wallis
- no comments
9) Results
- abbreviations of MSCs and WBCs are defined under Results (this is too late in the text).
- it is not clear how many samples have been used to analyse.
- more structure is needed under result; now sometimes confusing. The results should be rewritten (restructured).
- qPCR primers were used amplifying the 16S rRNA gene from all the germs present in the sample. Based on this technique, the conclusion is drawn that in somatic cells there is more bacterial DNA than in the white blood cells (see ... which indicates higher numbers of bacteria present in MSCs compared to WBCs).
This conclusion may also been drawn from the 454 sequencing data. After all, the average sequences in somatic cells is higher than in the white blood cells (see ... The mean number of sequences obtained from feces , milk somatic cells, and white blood cells were 8062 , 4164 and 3765 , respectively.)
It would have been more interesting to develop a qPCR for the genera found in all biological samples from the same animal to confirm the conclusion and also to screen the samples from the other animals because the qPCR is probably more sensitive than the 454 sequencing. Please comment.
10) Table 2
This table needs more explanation
Ruminococcaceae (is a family - 6.5%) <-> Ruminococcus (is the genus - 26,8%)
Bacteridales (is a order - 8,2%)
All others are families.
10) Discussion and conclusion
- see general comments
- line 220: why is the term milk leukocytes suddenly used? MSCs?
- in terms of physiology it is not correct to compare nursing women with machine milked cows.
- line 238: now it are milk cells...? see line 220
- line 245: it is known that these cannulas damage the teat canal. How do you know that there was no damage?
- external contamination will also result in phagocytosis by local micro- and macrophages. What about these bacteria?
- cellular composition of milk; use data of milk collected with same technique as yours. Dosogne collected with a similar technique (Dosogne et al. 2003 Differential Leukocyte Count Method for Bovine Low Somatic Cell Count Milk. Journal of Dairy Science, 86/3, 828–834).
- discussion on mechanisms of transport of DNA fragments; mention if homing has been demonstrated "in the bovine". In woman it is.
- near the end the discussion becomes extremely speculative. Claims have not been demonstrated in the current study.
Discussion and conclusion need more focus and more critical interpretation of own results. Advise to rewrite the discussion and the conclusion.
11) References
- I did not check the references

Experimental design

see basic reporting

Validity of the findings

see basic reporting

Additional comments

see basic reporting

---

## Round 0.2 · Minor Revisions

Although the quality of the manuscript improved significantly with this version, I would like to ask you to implement comments raised by both reviewers before accepting the manuscript for publication.

Reviewer 1 ·

Basic reporting

No additional comment

Experimental design

No additional comment

Validity of the findings

No additional comment

Additional comments

Lines 314-317 of the new version of the manuscript:Taxa found in common in all three environments ... Comments: these sentences should be completed with the full response provided by authors :
”The variation in the number of taxa found could be perhaps related to the stage of lactation, or when the cows were calved; or it could simply be from chance. In our study, we amplified very small quantities of bacterial DNA in the blood and milk, so sampling error could also come into play.”

Moreover, the following sentence: ”so sampling error could also come into play “ is inaccurate and should be rephrased as follows: “so it may be possible that bacterial species of interest were present in other studied samples but were not detected”.

Reviewer 2 ·

Basic reporting

Second peer
The quality of the paper improved significantly. The authors have to be congratulated for their efforts.

I still have some concerns:
1) regarding the focus on human milk that is presented here in this paper dealing with "cows milk". Again the physiology and the immunology of the cows udder is very different from the mammary gland of the human (the authors agree on that; see their answer in the discussion). I can accept that the authors refer to woman’s breast milk, however, they have to do it with much precaution.

45-52: human milk
53-56: the hypothesis of the authors is based on 45-52

AU: "We hypothesise that a similar mechanism of bacterial trafficking from the intestinal tract to the mammary gland, previously reported in lactating mothers, also occurs in the cow, contributing to the bacterial components found in raw cow’s milk and facilitating bacterial imprinting of the neonatal immune system.

REF: if this is the hypothesis of the authors I expect a clear answer in the conclusion. Is this hypothesis rejected or accepted on basis of the results?

256-262: The 3 papers that are cited refer (Martín et al., 2003; Beasley & Saris, 2004; Pérez et al., 2007 to potential sources of bacterial exposure for the breast-fed newborn (the human!!). This paper is dealing with cows milk.

If the authors have difficulties in finding "cow milk papers" I refer to the following recent paper that might be interesting to mention in the present publication:

Nathalie Bracke, Mario Van Poucke, Bram Baert, Evelien Wynendaele, Lobke De Bels, Wim Van Den Broeck, Luc Peelman, Christian Burvenich and Bart De Spiegeleer (2014) Identification of a microscopically selected microorganism in milk samples. J. Dairy Sci. 97 :609–615 http://dx.doi.org/10.3168/jds.2013-6932
ABSTRACT Identification of unwanted microbial contaminants microscopically observed in food products is challenging due to their low abundance in a complex matrix, quite often containing other microorganisms. Therefore, a selective identification method was developed using laser capture microdissection in combination with direct-captured cell PCR. This procedure was validated
with Geobacillus stearothermophilus and further used to identify microbial contaminants present in some industrial milk samples. The microscopically observed contaminants were identified as mainly Methylobacterium
species. Key words: laser capture microdissection , cultivationindependent
identification, bacterial contamination, direct-captured cell PCR

I am missing this cow paper; it needs your attention. LCM allows the selective isolation of cells under direct microscopic visualization. This provides is a new and unique technology that will
permit the study of the microbiome in milk of lactating cows.

2) REF: discussion on mechanisms of transport of DNA fragments; mention if homing has been demonstrated "in the bovine". In woman it is.
AU ANSW: To the best of our knowledge, the phenomenon of homing has not been shown in the cow.

REF: Why to their best knowledge when specific literature is available. Following paper also valid to be mentioned is:

J.A. Harp, P.L. Runnels and B.A. Pesch (1988) Lymphocyte Recirculation in Cattle: Patterns of Localization by Mammary and Mesenteric Lymph Node Lymphocytes
Veterinary Immunology and Immunopathology, 20 (1988) 31-39.
ABSTRACT
We examined patterns of lymphocyte localization in female dairy cattle following infusion of 51Cr-labeled autologous lymphocytes prepared from surgically excised mammary or ilea! mesenteric lymph nodes. Labeled lymphocytes prepared from mammary lymph nodes were recovered in proportionally high numbers from mammary and prescapular lymph nodes, and in low numbers from intestinal mesenteric nodes. This pattern was observed in both heifers and lactating cows. In contrast, labeled lymphocytes prepared from ilea! mesenteric lymph nodes of lactating cows were recovered in proportionally high numbers from intestinal mesenteric nodes, and in low numbers from mammary and prescapular nodes. These findings, when compared with previous results in sheep and swine, support the hypothesis that lymphocytes do not migrate efficiently between the gut and mammary gland of ruminants.

AND ALSO THE END OF THE DISCUSSION
Our previous results in a non-ruminant (Harp and Moon, 1988), in which mammary lymph node lymphocytes from swine showed preferential localiza­ tion in intestinal mesenteric nodes, are consistent with the hypothesis that a significant portion of the lymphocytes in the mammary nodes of the swine originated in the intestinal lymphoid tissue. This supports the presence of a functional entero-mammary link in swine, as suggested by previous studies (Evans et al., 1980; Kortbeek-Jacobs and Van der Donk, 1981; Salmon, 1986 ) . In contrast, mammary lymph node lymphocytes from sheep localized poorly in mesenteric lymph nodes, and accumulated preferentially in mammary and prescapular nodes (Harp and Moon, 1987). In the present study, this pattern of avoidance of intestinal nodes and preference for node of origin and peripheral nodes was seen in another ruminant species, the cow. Taken together, these data support the hypothesis that the entero-mammary link is less functional in ruminant species.

I am missing this cow paper; it needs your attention.

Experimental design

AU: Only cows identified as having a low "pooled quarter somatic cell count (SCC)", defined as 100,000 cells/mL.
REF: 1) This is "pooled milk" and this should not be considered as "free of pathogens". 2) The "rapid mastitis test" (Shoof International Ltd., Cambridge, NZ ) used to test SCC in individual milk. It is based on coagulation of the milk sample that indicates the presence of somatic cells. It is very difficult to accept that such an ordinary test (used by farmers; even far from being semi-quantitative; far from an accurate test, very rough) is used in the present study that claims such a fundamental finding. This is very difficult to accept in a paper publication that claims to be innovative. But to give the paper a chance I would advise to focus on this in the discussion. The authors would have used raditional identification methods for microbial contaminants that rely on phenotypic identification using cultivation-based methods and biochemical assays in individual milk that is analysed afterwards. Please be critical for all your techniques.
The statement that is mentioned in line 329 is false: "Cows used in the study showed no signs of intramammary infection"; how can you drop "such a hard statement" knowing that this is a simple non accurate farmer test???
See the technique explained in the paper of V.D. Bhatt, V.B. Ahir, P.G. Koringa, S.J. Jakhesara, D.N. Rank, D.S. Nauriyal, A.P. Kunjadia and C.G. Joshi (2012) Milk microbiome signatures of subclinical mastitis-affected cattle analysed by shotgun sequencing. Journal of Applied Microbiology 112, 639–650. "Quarter milk samples were subjected to cell counts using an electronic somatic cell counter (Foss, Hillerod, Denmark) and to bacteriological culture examination." I am missing this cow paper; it needs your attention.
Sample collection OK, near perfect.
Power analysis The authors should explain more in detail the method used to determine the number of animals to produce a significant effect (eventually with a reference).

Validity of the findings

Please be critical for the obtained results and "focus to milk of the cow". There is nothing wrong with this.

Additional comments

The paper is innovative and will receive positive comments at the condition that you are not afraid to recognise some flaws. If you don't you will receive a lot of negative comments by "cow experts". It is good to combat dogma's but do it with correct scientific data.

---

## Round 0.3 · accepted · Accept

Congratulations and thank you for your cooperation. I believe that your manuscript contains information, which might be interesting not only for experts in the field but also for the general scientific community.